# Neurochemical Alterations in Social Anxiety Disorder (SAD): A Systematic Review of Proton Magnetic Resonance Spectroscopic Studies

**DOI:** 10.3390/ijms23094754

**Published:** 2022-04-26

**Authors:** Sonja Elsaid, Dafna S. Rubin-Kahana, Stefan Kloiber, Sidney H. Kennedy, Sofia Chavez, Bernard Le Foll

**Affiliations:** 1Translational Addiction Research Laboratory, Campbell Family Mental Health Research Institute, Centre for Addiction and Mental Health, Toronto, ON M5S 2S1, Canada; sonja.elsaid@camh.ca (S.E.); dafna.kahana@camh.ca (D.S.R.-K.); 2Institute of Medical Science, Faculty of Medicine, University of Toronto, Toronto, ON M5S 1A8, Canada; stefan.kloiber@camh.ca (S.K.); sidney.kennedy@uhn.ca (S.H.K.); sofia.chavez@utoronto.ca (S.C.); 3Brain Health Imaging Centre, Centre for Addiction and Mental Health, Toronto, ON M5T 1R8, Canada; 4Department of Psychiatry, University of Toronto, Toronto, ON M5T 1R8, Canada; 5Campbell Family Mental Health Research Institute, Centre for Addiction and Mental Health, Toronto, ON M5T 1R8, Canada; 6Department of Pharmacology and Toxicology, Faculty of Medicine, University of Toronto, Toronto, ON M5S 1A8, Canada; 7Centre for Depression and Suicide Studies, Unity Health Toronto, Toronto, ON M5B 1M4, Canada; 8Li Ka Shing Knowledge Institute, Toronto, ON M5B 1T8, Canada; 9Krembil Research Institute, University Health Network, Toronto, ON M5T 0S8, Canada; 10Homewood Research Institute, Guelph, ON N1E 6K9, Canada; 11Departments of Family and Community Medicine, University of Toronto, Toronto, ON M5T 1R8, Canada; 12Addictions Division, Centre for Addiction and Mental Health, Toronto, ON M6J 1H3, Canada; 13Waypoint Research Institute, Waypoint Centre for Mental Health Care, Penetanguishene, ON L9M 1G3, Canada

**Keywords:** Social Anxiety Disorder, Social Phobia, Proton Magnetic Resonance Spectroscopy, Nuclear Magnetic Resonance, systematic review, neurochemicals, neurometabolites, neurotransmitters

## Abstract

(1) Objective: Considering that current knowledge of mechanisms involved in the molecular pathogenesis of Social Anxiety Disorder (SAD) is limited, we conducted a systematic review to evaluate cumulative data obtained by Proton Magnetic Resonance Spectroscopic (^1^H MRS) studies. (2) Methods: A computer-based literature search of Medline, EMBASE, PsycInfo, and ProQuest was performed. Only cross-sectional studies using ^1^H MRS techniques in participants with SAD and healthy controls (HCs) were selected. (3) Results: The search generated eight studies. The results indicated regional abnormalities in the ‘fear neurocircuitry’ in patients with SAD. The implicated regions included the anterior cingulate cortex (ACC), dorsomedial prefrontal cortex (dmPFC), dorsolateral prefrontal cortex (dlPFC), insula, occipital cortex (OC), as well as the subcortical regions, including the thalamus, caudate, and the putamen. (4) Conclusions: The evidence derived from eight studies suggests that possible pathophysiological mechanisms of SAD include impairments in the integrity and function of neurons and glial cells, including disturbances in energy metabolism, maintenance of phospholipid membranes, dysregulations of second messenger systems, and excitatory/inhibitory neurocircuitry. Conducting more cross-sectional studies with larger sample sizes is warranted given the limited evidence in this area of research.

## 1. Introduction

Social Anxiety Disorder (SAD) is the fourth most prevalent psychiatric disorder after Major Depressive Disorder, Specific Phobias, and Alcohol Use Disorder [1]. It affects 1.6–12.1% of people worldwide at least once in their lifetime [2]. Individuals with this disorder generally tend to fear and avoid social interactions and situations, which cause them significant disability in all aspects of life [1,2,3]. It is estimated that only 30% of those treated achieve a full recovery from SAD symptoms, warranting exploration of novel treatments [3,4]. Although disease mechanisms in SAD are not well understood, anatomical regions involved in ‘fear neurocircuitry’ are implicated [5,6].’ Fear neurocircuitry’ primarily includes thalamocortical, corticocortical, and corticostriatal pathways. These pathways include glutamate (Glu) neurotransmission, which is controlled by various mechanisms of feedforward, feedback inhibition, and disinhibition by several GABAergic inhibitory neurons [7].

Most sensory stimuli are routed via the thalamus and locus coeruleus (LC) to various cortical areas via their primary sensory source. There are multiple connections within the cortical areas and between the cortex and subcortical regions, allowing the assessment and interpretation of sensory stimuli and generating an appropriate behavioral response [8,9]. The amygdala appears to be the key player in the ‘fear neurocircuitry’ in the limbic system. In some pathological states such as SAD, the overactivity of the amygdala and insula may lead to ‘misinterpretation’ of ambiguous stimuli as a threat [9,10,11]. The insula also appears as an essential brain substrate in SAD that may participate in the ‘overinterpretation’ of familiar physical body sensations as stress responses that may secondarily initiate a fight or flight response via the periaqueductal gray (PAG) and hypothalamus [12,13,14]. In addition, the dorsal anterior cingulate cortex (dACC) and dorsomedial prefrontal cortex (dmPFC) may also contribute to the ‘misinterpretation’ of ambiguous sensory signals as threatening. From the dACC and dmPFC, the neural inputs to the rostral anterior cingulate cortex (rACC), ventromedial PFC (vmPFC), and orbitofrontal cortex (OFC) may not provide inadequate inhibitory inputs back to the amygdala [6]. Finally, hyperactivity of the caudate nucleus and of the putamen located in the striatum also plays a role in the ‘fear neurocircuitry.’ The consequences of the hyperactivity in these areas were previously linked to avoidant behaviors, often exhibited by individuals with SAD [15,16,17]. The anatomical brain regions thought to be involved in ‘fear neurocircuitry’ in SAD are displayed in Figure 1.

Proton Magnetic Resonance Spectroscopy (^1^H MRS) is a non-invasive, ionizing radiation-free imaging technique. It provides information on magnetic resonance signals originating from protons in the hydrogen nuclei of various molecules. ^1^H MRS signals can provide information on concentrations of neurometabolites that are byproducts of physiological processes and normal chemical metabolisms [18]. Thus, disturbances in these neurometabolites may point to aberrant disease mechanisms involving neurons and glial cells, which can be observed in neurological or psychiatric disorders. The neurochemicals that are studied using ^1^H MRS include N-acetylaspartate (NAA), total creatine (tCr), total choline (tCho), myo-inositol (mI), Glu, glutamine (Gln), glutamate + glutamine (Glx), and γ-aminobutyric acid (GABA). Specifically, dysregulation of these metabolites indicates a disturbance in the neurons and glial cells [19]. In this systematic review, we describe each examined metabolite and how they can be linked to *specific molecular mechanisms within neurons and glial cells*.

N-acetyl-aspartate (NAA) is an amino acid highly concentrated in the nervous system at 6–12 mM [20]. It is also a vital brain osmolyte that increases protein stability at physiological pH [21,22,23]. NAA is synthesized in neuronal mitochondria from the two byproducts of glycolysis (aspartate and acetyl-CoA) [20]. Disturbances in neuronal energy consumption in mitochondria may lead to inadequate production of NAA, which in ^1^H MRS may be detected as NAA downregulation.

From the neuron, NAA is directly transported to oligodendrocytes [24], where it is hydrolyzed by the enzyme aspartoacylase (ASPA) into L-aspartate and acetate [25]. Acetate is an essential component of lipid synthesis and eventual myelin formation. Consequently, impaired ASPA may lead to increased levels of NAA in oligodendrocytes, accompanied by inadequate myelin production [26,27].

It is estimated that Cho concentration ranges from 1 to 2 mM. In ^1^H MRS, the signal from Cho, designated as total choline (tCho), reflects not only the amount of cytosolic Cho but also the amount of phosphocholine (PC) and glycerophosphocholine (GCP) [28,29]. tCho metabolites are predominantly involved in phospholipid membrane metabolism, and they cycle from intracellular free Cho via PC and GPC into phosphatidylcholine (PYC). Pathological conditions contributing to elevated cellular membrane breakdown (e.g., necrosis) escalate the decomposition of PYC back into GPC, PC, and Cho. In ^1^H MRS, this change is detected by increased levels of tCho. Alternatively, the mechanisms of cellular membrane repair utilizing Cho compounds to rebuild the phospholipid bilayer can be detected by decreased levels of tCho [30].

Creatine (Cr) is a nitrogen-containing amino acid. It has an essential role in regenerating ATP, maintaining membrane potentials, ion gradients, calcium homeostasis, and scavenging reactive oxygen species (ROS) [31,32]. The concentration of Cr in the human brain is estimated to be anywhere between 7.5 mM and 11.1 mM [18,33]. In ^1^H MRS, total creatine (tCr) estimates the sum of both Cr and phosphocreatine (PCr) [33,34].

tCr is synthesized in neuronal and glial cells. Peripheral tCr is transported through the blood-brain barrier to the brain. Once in the brain, tCr enters a Cr/PCr/creatine kinase (CK) system, serving in ATP regeneration [35,36]. Cellular depletion of ATP is marked by decreases in tCr levels, indicating aberrant mitochondrial function [33].

Myo-Inositol (mI) is one of the most biologically abundant isomers of glucose (~6 mM) found in the brain [37]. It is mainly synthesized in neurons but is stored in glial cells [38]. In neurons, mI is an essential precursor of the G-protein-coupled phosphoinositol (PI) second messenger system, which is critical in forming membrane phospholipids. Increases in cellular membrane turnover may lead to higher mI levels than expected. Psychiatric conditions (including SAD) may impact the downregulation of neurotransmitters (e.g., serotonin, dopamine, glutamate, and acetylcholine) that bind to such G-protein-coupled receptors. Consequently, these neurotransmitter changes may manifest levels of mI that are lower than normal [39,40,41,42].

Glutamate (Glu) is a major excitatory neurotransmitter that plays an essential role in cognition, learning, memory, regulation of neuroendocrine secretions, and neuroplasticity of synaptic connections. The concentration of Glu ranges between 6 and 12.5 mM, and its largest pools are found in glutamatergic neurons. Glutamine (Gln) is an amino acid with a primary role to serve as an intermediary metabolite in the Glu–Gln neurotransmitter cycle. Gln is mainly found in astrocytes, and its concentration ranges between 2 and 4 mM. Glu–Gln inter-conversion reflects the interaction between neurons and astrocytes [18,43].

Glu and Gln are the markers of neuron–astrocyte integrity. Injury to astrocytes and not to neurons may upregulate Glu/Gln ratios, ultimately causing excitatory/inhibitory neurotransmission disturbances. Alternatively, specific injuries to neurons, but not to astrocytes, may indicate elevated Gln, but not Glu, whereas impaired integrity of both neurons and astrocytes may be detected by decreased Glu and Gln [44,45]. Given that Glu and Gln ^1^H MRS signatures are very much overlapping due to similarity in molecular structures, at lower field strengths (≤3T), they cannot be separated. Consequently, ^1^H MRS concentrations of these two neurometabolites are often reported together as glutamix (Glx) at these lower magnetic field strengths. Glx also contains trace amounts of γ-aminobutyric acid (GABA) [17,46].

Gamma-aminobutyric acid (GABA) is a major inhibitory neurotransmitter in the mammalian brain. It is present at small concentrations (~1 mM) [18,47]. GABA is synthesized by decarboxylation of Glu by two isozymes of glutamic acid decarboxylase (GAD) 65 and GAD 67. It binds to ionotropic GABA_A_ and metabotropic GABA_B_ receptors, which are important in inhibitory neuromodulation of brain currents [45,48,49]. Acute and chronic stress were observed to decrease the function of GAD 67, and impede GABA release from the astrocytes, ultimately leading to the dysregulation of the inhibitory currents at the post-synaptic neurons [50].

In ^1^H MRS, GABA resonates at 3.01 ppm, 2.28 ppm, and 1.89 ppm [18]. Given that more intense signals resonating from other metabolites at similar frequencies obscure GABA peaks (e.g., peaks at NAA at 2 ppm, Cr at 3 ppm, and Glx at 2.3 ppm), GABA cannot be measured using a standard PRESS or STEAM sequence at magnetic fields < 4 T. Alternatively, special spectral editing methods are needed to ‘unravel’ GABA resonance. The most commonly used spectral editing methods are J-edited PRESS-based methods, such as MEGA-PRESS and double-quantum filtered (DQF) [51].

The purpose of this systematic review is to evaluate metabolic differences associated with SAD. Explicitly, this review aims to investigate which neuronal and glial cell molecular mechanisms of action (as described by disturbances in levels of spectroscopic metabolites) are suggested for the pathogenesis of SAD.

## 2. Methods

### 2.1. Search Strategy

Using the OVID platform (OVID Technological Inc.), a computer-based literature search of Medline, EMBASE, and PsycInfo was conducted for all published articles between 1 January 1946 and 20 January 2022. The search was conducted according to PRISMA (Preferred Reporting Items for Systematic Reviews and Meta-Analysis) criteria [52]. The search keywords used were: ‘Proton Magnetic Resonance Spectroscopy,’ OR ‘Nuclear Magnetic Resonance,’ AND ‘Social Anxiety Disorder,’ OR ‘Social Phobia.’ During pre-screening, original articles written in English that examined populations with SAD versus HCs were only included.

The identified publications were imported into EndNote X9 (Clarivate Analytics), and duplicates were removed. Subsequently, titles and abstracts were manually scoped. Only human in vivo ^1^H MRS studies were selected in which participants with SAD were compared to HCs without any history of psychiatric disorders. The screening assessment identified articles that were thoroughly read and assessed for eligibility. During the screening process, the exclusion criteria were the following: (I) review articles, (II) editorials/commentaries, (III) additional duplicates, and (IV) off-topic articles, which did not include cross-sectional ^1^H MRS studies comparing individuals with SAD and HCs. We also conducted a search for on-topic theses and dissertations, using ProQuest. Two authors (S.E. and D.S.R.-K.) conducted the entire systematic search separately. Following the individual examination of the literature, both authors discussed and agreed upon the search results.

### 2.2. Search Selection

The initial computer-based search generated 469 studies. The search of the ProQuest database yielded another document (a thesis) on the topic of interest. The identification process involved removing the duplicate articles (*n* = 24), leaving 446 articles to be screened by title and abstract. During the screening process, an additional 428 articles was excluded. Among the excluded, 350 were off-topic, 67 were review articles, 7 were additional duplicates, and 4 were commentaries/editorials. Finally, 18 articles were sought for retrieval; however, 3 could not be obtained, leaving 15 to be assessed for eligibility. Of those, 7 were excluded because either spectroscopy techniques were not used (*n* = 4), SAD patient population was not studied (*n* = 1), fluoride MRS was used instead of ^1^H MRS (*n* = 1), or cross-sectional comparisons between participants with SAD and HCs were not made (*n* = 1). Only 8 studies were finally included in this qualitative synthesis [53,54,55,56,57,58,59,60]. The results of the systematic search are displayed in Figure 2.

### 2.3. Data Extraction

Table 1 and Table 2 synthesize information extracted from the selected studies that was initially imported into the standardized Microsoft Excel spreadsheet (Redmond, WA, USA). The information extracted from each study included names of authors; years of publication; demographic characteristics of enrolled participants, including the number of participants, their sex, age, the severity of SAD (described by LSAS scores), type of psychiatric comorbidities, and their medication status; and whether matching according to age–sex was performed between the SAD and HC groups. Information on spectroscopic methodologies was also derived, including the brand of MRI scanner, magnetic field, ^1^H MRS method, sequence and sequence parameters, examined ROIs, voxel sizes, and a list of neurometabolites studied for each region of interest (ROI). The means, standard deviations, statistics, and *p*-values were obtained from each cross-sectional comparison (SAD vs. HCs) for each metabolite per region. Moreover, information from two studies that tested the effects of treatments was obtained and is described in the Results section.

## 3. Results and Discussion

### 3.1. Results

Eight identified studies assessed 146 individuals with SAD and 176 HCs [53,54,55,56,57,58,59,60]. The demographic characteristics of the participants are presented in Table 1. Structured Clinical Interview for Diagnostic and Statistical Manual (DSM) for Mental Disorders 5 (SCID-5) was used to determine the diagnosis of SAD and the lack thereof in HCs. Generally, smaller sample sizes ranging from 9 to 36 SAD participants were used. SAD participants and HCs were matched according to sex and age in six studies [53,55,56,57,59,60]. Five studies [55,59,60] also enrolled participants with other comorbidities in the SAD group.

In contrast to other studies, Grills employed two- and three-group comparisons of metabolites. The three groups that were compared were: (1) participants with only SAD, (2) participants with SAD and Major Depressive Disorder (MDD), and (3) HCs without a prior psychiatric history. In the two-group comparison, the first two groups (SAD only and SAD + MDD) were combined and compared to the HCs. Metabolite differences between male and female participants were assessed for select metabolites in only two of the eight studies: Glu and NAA in the study by Grills [58], and Cho and mI in the study by Tupler et al. [54].

In addition, two studies assessed the effects of pharmacological treatments on metabolite changes in participants with SAD [53,57]. Seven studies used the Liebowitz Social Anxiety Scale (LSAS) to identify the severity of illness [54,55,56,57,58], whereas the study by Davidson et al. utilized Brief Social Phobia Scale (BSPS) instead [53]. LSAS total scores varied across studies from 57.3 ± 11.5 [59] to 88.6 ± 24.82 [60].

Table 1 summarizes the ^1^H MRS methodology described in the studies. Neurometabolites were assessed with MRI scanners at magnetic fields ranging from 1.5 T to 4 T [53,54,55,56,57,58,59,60]. Single-voxel spectroscopy (SVS) was used more commonly when scanning cortical brain regions [55,56,58,59,60]. Two-Dimensional Chemical Shift Imaging (2D CSI) was utilized to measure smaller sub-cortical structures [53,54,57,60], given that 2D CSI has an increased signal relative to SVS and can, thus, measure metabolite levels in smaller structures.

The two ^1^H MRS methods, STEAM (stimulated echo acquisition mode) and PRESS (point resolved spectroscopy) were used for determining the levels of NAA, Glx, Cho, mI, and Cr [53,54,55,56,58,59,60]. PRESS offers a better signal-to-noise (SNR) ratio and is less sensitive to motion, whereas STEAM may be more effective for measuring metabolites with short T2 decays because it can be performed with shorter echo times [61]. GABA was measured using the MEGA-PRESS sequence, which uses the J-editing technique [57], and PRESS with the Double-Quantum and Filter-Selective Editing Technique for Refocusing GABA Signal (DQF-S) [58]. The selected studies examined seven regions of the brain known to be functionally aberrant in individuals with SAD: the thalamus, ACC, insula, dlPFC, caudate, putamen, and the occipital cortex. Moreover, two studies examined larger ROIs that encompassed several brain structures (please see Table 1 for details).

Below, we report the differences in relative metabolite concentrations between participants with SAD and HCs for each ROI. A more detailed inventory of study results (including means, standard deviations, and *p*-values) is presented in Table 2.

Seven studies provided NAA measurements. Significantly higher NAA was found in SAD participants’ left insula, dlPFC, and the right thalamus [55,59,60]. Conversely, in SAD, lower NAA was found in the OC [56], in subcortical white, and two gray matter voxels (one containing mostly thalamus and the other containing caudate and parts of the cortical gray matter) [54]. Decreased NAA/Cho was observed in the SAD group [53].

Mixed results were observed when studying the ACC. The two-group comparison by Grills revealed reduced NAA/H_2_O in females with SAD, whereas no differences were found in male participants [58]. In contrast, the three-group comparison showed no significant effects of sex or diagnosis on NAA/H_2_O [58]. Two other studies discovered elevated NAA/tCr in the ACC in individuals with social anxiety [55,56].

Moreover, a positive correlation between NAA/tCr and LSAS scores was found in the insula [55], ACC [56], and cortical gray matter [54]. Contrary to these findings, a negative correlation was observed between NAA/tCr and total LSAS scores in the OC [56]. NAA/tCr did not differ between SAD participants and HCs in the left and right caudate and putamen [55,59,60]. In addition, brain slices that included the thalamus, caudate, and putamen (designated as the subcortical gray matter) indicated no difference in levels of NAA/Cr [54].

Six out of eight studies reported on tCho resonance. No significant differences between SAD participants and HCs were reported in studies examining the left and right thalamus, caudate, putamen [59,60], voxel-containing sub-cortical gray matter [54], OC [56], dlPFC [59], and insula [55], whereas in ACC, three out of four studies also observed no difference in tCho/tCr between SAD and HC groups [55,59,60].

Decreased tCho/tCr was noted in the ACC [56] and subcortical voxel where tCho SNR was reported [53], whereas elevated tCho/tCr was recorded in the cortical gray matter [54]. tCho/tCr is positively correlated with LSAS scores in the left thalamus [60], whereas a negative correlation between these two variables was found in the subcortical gray matter [54]. A negative correlation was found in subcortical gray matter (thalamus) between tCho SNR and BSPS [53].

Overall, tCr was evaluated in three studies. No changes were observed in the ACC [56,59], OC [56], thalamus, the left or right putamen [59], and white matter (reported as tCr SNR) [53]. In contrast, significantly lower tCr was seen in the dlPFC [59], subcortical gray matter, and the voxel with cortical and subcortical gray matter [53] in individuals with SAD. Moreover, in this study, tCr levels are negatively correlated with the LSAS scores in the dlPFC [59].

In total, five studies assessed mI in individuals with SAD and HCs. No significant differences between groups were observed in the insula [55], OC [56], and the white matter [54]. Upon the examinations of ACC, three studies reported no difference in mI/tCr between SAD and HC groups [54,55,60], whereas the study by Grills demonstrated decreases in mI/H_2_O in the combined SAD group and the SAD group alone, as well as in the SAD + MDD group compared to the HCs in dmPFC/ACC [54,55,58,60]. Conversely, cortical and subcortical gray matter scans indicated significantly higher mI/tCr and mI/NAA in SAD participants. Moreover, mI/tCr is negatively correlated with total LSAS scores in subcortical gray matter in the SAD group [54].

Glutamate levels were measured in four studies. No statistically significant differences between HCs and SAD were observed in the OC [56], thalamus [57], and female participants in the dlPFC/ACC [58]. Glu was reported as being lower and higher in the ACC in the SAD group in two separate studies [56,60]. Moreover, increased Glu was observed in the whole brain [57] and in dmPFC/ACC, but only between male participants [58]. A separate analysis of Gln was conducted on the whole brain and thalamus, in which Gln was significantly higher in the SAD group [57]. The only significant correlation between Glu levels and total LSAS scores in the SAD group was in the ACC, and it was reported as positive [56].

Two studies reported on levels of GABA in individuals with SAD vs. the HCs. Although no between-group differences were observed in the whole brain, decreased, GABA/tCr was recorded in the SAD group in the thalamus [57]. In the study by Grills, the three-group comparison revealed significantly lower GABA/H_2_O in the dmPFC/ACC in the SAD group but not in the SAD + MDD group compared to the HCs. However, no differences in GABA levels were reported in the two-group comparison between the combined SAD groups and the HCs [58].

In addition to cross-sectional comparisons, only two out of eight studies examined the effects of SAD treatments on metabolite changes in the SAD group. The study by Davidson et al. administered clonazepam (a benzodiazepine customarily used to treat acute anxiety) to three participants with SAD for ten weeks. However, due to an insufficient number of participants treated, no specific conclusions on the effects of clonazepam could be made at the time [53].

In the second study, eight-week treatment of 10 SAD participants, each with an average dose of 2100 mg/day of the antiepileptic drug levetiracetam, significantly decreased Gln/tCr in the thalamus. No post-treatment changes were observed in GABA/tCr, or Glu/tCr in either brain region studied [57].

### 3.2. Discussion

To our knowledge, this is the first review synthesizing what is currently known about ^1^H MRS metabolite differences between SAD and HCs and the pathophysiological mechanisms of SAD. However, considering that most studies did not have sufficient statistical power to show the effect of SAD on tissues in the brain, our discussion section focuses mainly on interpreting results from studies in which the statistical power was nearly achieved. The findings from this systematic review are discussed based on the questions posed in the subtitle of each section.

#### 3.2.1. What Are the Metabolite Differences Associated with SAD?

NAA is the most-studied metabolite; however, upon closer inspection of the quality of evidence, several observations should be noted. Firstly, only a handful of NAA comparisons achieve a statistical power of ~80%, sufficient for mitigating Type II error. The studies with a higher power showed upregulations in the NAA in the ACC [55], left insula [55], dlPFC [59], and the right thalamus [60]. The downregulated NAA was noted in the OC [56], in the dmPFC/ACC (but only females) [58], and in three broader voxels with gray and white matter [53].

Secondly, sufficient statistical power is ‘more easily’ achieved in the cortical (ACC, dlPFC, insula, and OC) rather than smaller subcortical regions (caudate, putamen, thalamus). Consequently, for those smaller regions, much larger sample sizes are needed to demonstrate the effect of SAD on NAA.

Thirdly, in six out of eight studies, researchers matched their participants according to age and sex [53,55,56,57,59,60]. As the levels of NAA were shown to vary accordingly [62], age and sex differences between groups may have confounded NAA results in each group in the two studies that did not perform age and sex matching [54,58]. Fourthly, the downregulated NAA in the study by Davidson et al. was reported in terms of SNR and not in terms of metabolite concentrations. Metabolite SNRs may be proportional to metabolite concentrations, yet they are not entirely equivalent. SNR is a measure of the signal relative to the standard deviation of the noise; as such, it reflects the data quality more than the concentration [63]. Alternatively, metabolite concentrations are reported as ratios to internal reference metabolites (tCr) to counteract the inter-subject anatomical, voxel-tissue-specific differences [64,65]. Although NAA SNRs in subcortical tissues of SAD were decreased compared to the HCs, no differences were observed in NAA/tCr between the groups in this comparison [53].

tCho was the second most examined metabolite across various cortical and subcortical regions. The comparisons that achieved a power of >80% only included studies in the cortical gray matter (increased tCho in SAD vs. HCs), [54], right putamen (decreased tCho in the SAD group) [59], and two sub-cortical gray matter voxels, indicating lower tCho SNR in the SAD group, but no statistically significant differences in the subcortical white matter [53].

Phan et al. also reported decreased Cho/tCr in the ACC, but presumably, due to the low sample size of 10 participants in each group, their comparison only demonstrated the power of 69% [56]. Moreover, this study suggested that, as the severity of SAD symptomatology increases, tCho tends to be downregulated in the ACC.

Contradicting results from the cortical voxels may have culminated from differences in the compositions of investigated regions. For instance, Tupler et al. [54] studied broader cortical regions, including the parts of the temporal and frontal lobes, whereas Phan et al. [56] explicitly focused on the ACC. It is not currently clear whether all neurocircuits in the frontal and temporal lobes combined have a role in the pathogenesis of SAD; thus, the presented evidence is still equivocal and needs to be further investigated.

Several observations also can be noted about the demographic characteristics of participants enrolled in the study by Tupler et al. that may have impacted the study’s findings. Fourteen out of 19 participants had social anxiety, but only 4 out of 10 HCs were female. In the same study, Cho/tCr was higher in male participants, which may have introduced a bias towards reporting tCho lower than expected in the SAD group. Differences in age between SAD and HC groups may have impacted the study findings, given that tCho/tCr is positively correlated with age. Moreover, in the study by Tupler et al., the SAD group was slightly older (mean age 42) than the HCs (mean age 38), possibly suggesting that higher-than-expected tCho levels may have been noted in the SAD group [54].

In summary, the evidence of tCho in SAD groups is not clear, as the impact of demographic characteristics is not evident. Nevertheless, the findings of tCho in the SAD group suggest neuronal and glial cell abnormalities in the membrane turnover, which may be more likely to occur in gray matter that includes neuronal cell bodies and glial cells as well as unmyelinated rather than myelinated axons.

Lower tCr levels were observed in all comparisons of participants with SAD and HCs; however, tCr was only significantly downregulated in the left dlPFC [59] along with two voxels containing mostly gray matter, investigated by Davidson et al. [53]. However, this study only reported tCr SNRs; thus, the effect of SAD on metabolite concentrations can only be speculated. Consequently, although the current evidence is limited, it may indicate that SAD pathogenesis is associated with impairments in mitochondrial energy metabolism [66].

The accumulated evidence on mI in SAD is minimal and inconsistent. Only two studies had adequate statistical power (~90%). Accordingly, the increases in mI were noted in the cortical and subcortical gray matter [54], whereas downregulated mI was found in the dmPFC/ACC [58]. From the mI analysis in the smaller subcortical voxels (i.e., caudate and putamen), much larger sample sizes were needed to demonstrate the effect of SAD on changes in mI (~40 for caudate and 200 for the putamen) [55]. Nevertheless, both mechanisms of mI upregulation indicating cell membrane breakdown and mI downregulation pointing to decreased activity of PI second messenger systems and cell membrane repair may be involved in the pathogenesis of SAD.

Two comparisons of Gln in the whole brain and thalamus displayed increases in the SAD group with an adequate effect size (Cohen’s d > 1.4) [57]. However, for Glu, only three out of six studies had at least 70% statistical power. Increased levels of Glu in the SAD were noted in the ACC [56] and the whole brain [57]. In contrast, decreased Glu was also recorded in the ACC [60], whereas no differences in Glu were recorded in the thalamus [57]. However, it may be reasonable to deduce that both increases and decreases in Glu, separately, may be responsible for the pathogenesis of SAD, given that demographic characteristics were similar in both studies. The two comparisons with adequate statistical power showed decreases in GABA in the thalamus [57] and dlPFC/ACC [58] in the SAD groups. Cumulatively, Glu, Gln, and GABA dysregulation imply impairment in Glu–Gln, and Glu–GABA interconversion that can be directly linked to imbalances in E/I glutamatergic neurocircuitry.

#### 3.2.2. What Molecular Mechanisms Were Implicated for SAD?

Several specific molecular mechanisms demonstrating neuronal and glial cell appearance have been implicated for SAD. We have proposed these mechanisms in Figure 3 based on the accumulated evidence from the eight spectroscopic studies reviewed here. In this figure, the mechanisms displayed in light gray indicate mechanisms that ^1^H MRS studies have implicated.

Lower levels of NAA (in ACC and OC), and tCr (in subcortical gray matter and dlPFC), may be linked to *mitochondrial dysfunction* and disturbances in energy metabolism [67]. Direct injury to neural mitochondria impairs the synthesis of NAA, which in ^1^H MRS is detected by lower-than-expected levels of NAA [19]. Furthermore, injury to mitochondria leads to downregulation in oxidative phosphorylation responsible for ATP synthesis. Decreased tCr often results from compensatory mechanisms involving increased activity of CK that utilizes tCr to produce ATP [33].

The evidence supporting mitochondrial dysfunction and impaired energy metabolism in SAD comes from animal studies. In one study, cuprizone-fed C57Bl6 mice (mitochondrial toxin and ROS generator) revealed increases in animal behaviors of anxiety associated with impairments in neuronal mitochondria in the mPFC and decreases in NAA [68,69]. In another study, mitochondrial dysfunction was induced by early socially isolated mice compared to their socially conditioned controls. The socially isolated mice also displayed elevated levels of ROS and lower levels of ATP (both indicating mitochondria dysfunction)] in the brain [70]. Moreover, a 15-day tCr supplementation to pentylenetetrazole-treated mice that previously exhibited anxiety demonstrated anxiolytic effects [71].

Higher-than-expected levels of NAA (as seen in ACC, insula, dlPFC, and thalamus) may result from damage to oligodendrocytes, leading to impaired ASPA in the oligodendrocytes and ultimately to *inadequate myelin production* [19]. For instance, in a study in which Balb/c mice exposed to social defeat stress had elevated NAA in the brain three weeks after exposure, NAA increased concurrently with reduced myelin basic protein (MBP) immunoreactivity, indicating impeded myelination [22].

Higher-than-usual tCho and mI levels were observed in SAD participants’ cortical and subcortical gray. On the other hand, lower tCho and mI were reported in the ACC, suggesting two proposed mechanisms of SAD-induced toxicity: *neuronal and glial cell membrane breakdown and repair*. These mechanisms are not mutually exclusive but instead may occur in consecutive order. Initially, neuroinflammatory processes lead to the breakdown of integral parts of the cell membrane, such as PYC into tCho or abnormally upregulated phosphatidylinositol compounds, leading to higher mI [54]. Following cellular membrane breakdown, cell membrane repair compensating mechanisms are activated, ultimately leading to the utilization of free-flowing compounds such as tCho and mI to rebuild the phospholipid bilayer [72]. Thus, both mechanisms may occur depending on the stage of the molecular pathogenesis of SAD.

The evidence from human and animal studies indicates that both increases and decreases in tCho and mI may be linked to increased anxiety. For instance, compared to HCs, in separate spectroscopic studies, participants with SAD had increased tCho in the dmPFC study by Raparia et al. [73], whereas in the study by Moon and Jeong, decreased tCho was noted in the dlPFC [74]. In animal studies, anxiety-like behaviors increased mI, and gliosis in Wistar rats injected with a neuroinflammatory agent was reversed when a non-steroidal anti-inflammatory agent was administered [75]. Conversely, in the elevated plus-maze test, sodium-dependent myo-inositol cotransporter-1 heterozygous knockout mice with 15% less mI in the frontal cortex showed anxiety-like behaviors [76].

Another potential mechanism of action that could result in lower mI in patients with SAD is auto-oxidation of neurotransmitters, which may be perpetuated by oxidative stress in neurons and glial cells during anxiety states [77,78]. Accordingly, ROS damage these neurotransmitters, rendering them inactive. For instance, dopamine reacts with ROS, giving rise to the quinone form and superoxide anion, the latter of which further decomposes into various ROS. In this state, ‘the decomposed dopamine’ cannot efficiently bind to its receptors [78]. Given that neurotransmitters such as dopamine bind to G-protein-coupled receptors that utilize PI second messenger systems (including inositol triphosphate (IP_3_) and diacylglycerol (DAG)), it is not surprising to observe that lower mI levels may occur as a consequence of *the lower activity of PI second messenger systems* and lower levels of these neurotransmitters [54,75,76,77,78,79]. It is noteworthy that lower levels of dopamine and serotonin were previously linked to the pathophysiology of SAD [78,79,80].

Lastly, *impairments of Glu–Gln, and Glu–GABA interconversion*, which may be directly linked *to the imbalances in E/I glutamatergic neurocircuitry*, were also implied in ^1^H MRS studies reviewed in this article. In SAD groups, higher Gln, lower GABA, and unchanged Glu in the thalamus may indicate impaired activity of neurons and not astrocytes, leading to inadequate Gln reuptake and extracellular accumulation [57]. Likewise, damage to GABA interneurons, specifically to GAD and GABA transporters, may have led to decreases in GABA in SAD groups [45,50,57]. Furthermore, when GABA is depleted, the activity of GABA on GABA receptors in post-synaptic neurons may be insufficient for dampening excitatory currents, leading to imbalances in glutamatergic neurocircuitry in favor of overexcitation. Similarly, upregulated Glu and Gln and unchanged GABA in the whole brain may demonstrate imbalances in E/I neurocircuitry in individuals with SAD [57].

In the ACC, injury to astrocytes and neurons (↓ Glu) [60], astrocytes only (↑Glu) [56,58], and GABA interneurons (↓ GABA) [58] may have occurred in those with SAD, and these metabolic imbalances may have contributed to aberrant glutamatergic neurocircuitry. The evidence from animal models of SAD shows that increased levels of Glu were observed in pyramidal neurons in the PFC, lending support that both the upregulation and downregulation of Glu may occur separately in the PFC in those affected by SAD [81]. Alternately, adolescent Balb/c mice exposed to social defeat stress exhibited lower Glx [22].

In addition, animal models of social anxiety (using social fear conditioning) focused on examining *the imbalances in E/I glutamatergic neurocircuitry* demonstrated increased glutamatergic neurotransmission and decreased activity of GABA interneurons in the dmPFC, ACC, and amygdala [44,81,82,83]. Conversely, compensatory activation of parvalbumin (PV) GABAergic interneurons in the dmPFC increased social interaction in SFC mice [44,81,84], whereas the administration of muscimol (the GABA_A_ receptor agonist) ameliorated previously observed social deficits. Taken together, these studies support the importance of Glu, Gln, and GABA in the pathophysiology of SAD.

#### 3.2.3. Which Other SAD Molecular Mechanisms Should Be Investigated in Human ^1^H MRS Studies?

Although the current evidence from eight ^1^H MRS studies implies injury to various neuronal and glial cell organelles, the overarching mechanism (oxidative stress) potentially responsible for cellular damage has not been explicitly studied. Notably, during stress, the hypothalamic–pituitary–adrenal (HPA) axis stimulates the release of glucocorticoids, which, when active on a long-term basis (such as in SAD), may lead to oxidative damage [77,85]. In individuals with SAD, the occurrence of oxidative stress is demonstrated by increased levels of malondialdehyde (MDA) (a product of lipid peroxidation), and antioxidant enzymes typically activated during oxidative stress (such as superoxide dismutase (SOD), catalase (CAT), and glutathione peroxidase (GPx)) [86,87]. Consequently, future ^1^H MRS studies should examine the presence of oxidative stress in various brain regions by directly measuring biomarkers, such as glutathione, taurine, or ascorbate, which may be observed at lower-than-usual concentrations during oxidative stress.

Another pathological mechanism implied but not directly measured is higher glycolysis. As impairments in mitochondrial oxidative phosphorylation that activate ATP compensatory reactions such as glycolysis have been implicated in SAD, it would be interesting to measure lactate levels. Lactate is the byproduct of glycolysis; thus, increases in the rates of glycolysis would be noted by higher-than-expected lactate levels. Unlike tCr, lactate is the direct measure of glycolysis [33].

#### 3.2.4. What Does the Evidence from ^1^H MRS Studies Imply about the ‘Fear Neurocircuitry’?

Given that ‘fear neurocircuits’ primarily communicate via neurotransmission of Glu and GABA, the dysregulation of Glu–Gln and Glu–GABA cycling (as described above) may be directly linked to cellular perturbances observed in E/I neurocircuitry. However, it is currently unclear how imbalances in other ^1^H MRS metabolites in SAD relate to the overactivity of Glu/Gln and/or downregulated GABA. The specific correlation between Glu or GABA levels and other metabolites should be explicitly studied in SAD to understand this topic better.

Such comparisons of ^1^H MRS metabolites, for example, have been conducted in ^1^H MRS studies involving antipsychotic naïve patients with first-episode psychosis (FEP) [88]. The researchers showed that Glu is positively correlated to Cho and mI in the FEP group. Considering that elevated Cho and mI in cellular levels may indicate neuroinflammatory processes in astrocytes, the authors concluded that astrocyte dysfunction may have led to overaccumulation of Glu [88]. In another ^1^H MRS study involving remitted patients with Bipolar Affective Disorder I, a strong correlation between Glu and NAA was observed in the hippocampus. It was, moreover, indicated that NAA served as a reservoir for Glu by converting into Glu on demand [89]. The association between tCr and E/I in fear neurocircuitry was noted in animal studies. These experiments demonstrate that tCr acts as a partial agonist for GABA_A_ receptors; thus, depleted tCr may result from downregulated inhibitory neurocircuitry [90,91]. Furthermore, Cr supplementation dampened extracellular Glu levels in Huntington’s disease (HD) [92]. Consequently, future ^1^H MRS studies in SAD should investigate how tCho, NAA, and mI correlate with Glu and how tCr correlates with Glu and GABA.

#### 3.2.5. What Are the Study Limitations Discussed in this Review?

High heterogeneity of data and only eight studies investigating each metabolite per region made performing meta-analysis impractical. One of the contributing factors to data heterogeneity may be that each study employed different MRS techniques. Three studies utilized a low magnetic field (1.5T) to measure metabolic differences in SAD [53,54,55]. Due to increases in the peak overlaps and limited SNR at 1.5T, readings of spectroscopic metabolites may not have been precise. Earlier studies by Davidson et al. and Tupler et al. used larger ^1^H MRS voxels encompassing several brain regions (e.g., cortical gray matter including the prefrontal and temporal cortex) [53,54]. Although larger voxels may produce better SNR, individual contributions from each anatomical region (within a larger voxel) to the metabolite readings may not have been apparent.

Moreover, newer processing techniques use larger basis sets, allowing for the measurement of more metabolites. In an earlier study by Davidson et al., the peak areas and SNRs were extracted [53]. This method only permits the measurement of metabolites resonating with large single-peak intensities, such as NAA, Cho, and Cr. The latter studies used peak fitting, whereas current studies use more elaborate and accurate methods relying on basis sets, including the full spectral signatures of metabolites. LC Model is a software tool that is commonly used, as it allows the quantification of several metabolites simultaneously by solving the linear combination of model spectra in an automated fashion with no subjective user input.

Different ^1^H MRS methods were used across eight studies. SVS was more likely to be used when measuring larger cortical regions, such as ACC or dlPFC [55,56,58,59]. SVS allows for better localization and shimming than 2D CSI, while 2D CSI can measure metabolites in smaller volumes, such as those required for the caudate, putamen, or thalamus [57,60,93,94]. To further complicate things, different spectral sequences and timing parameters were employed. The STEAM sequence typically allows better quantification of more rapidly decaying metabolites, whereas higher SNR is usually achieved using PRESS [61]. Even when measuring the same ROIs, voxel volume differences may have also contributed to the variability of study results. Larger voxels generally produce better SNRs, thus allowing for more accurate measurements of metabolites [65].

Another limiting factor is how metabolites were reported in each study. As previously mentioned, Davidson et al. mainly reported metabolite SNRs [53], whereas most other studies noted their metabolites in reference to tCr [54,55,56,57,59,60]. A study by Grills and Yue et al. stated metabolite measurements relative to the internal water concentrations [58,59]. Reporting metabolites/tCr may pose a problem because downregulated tCr (as observed in the dlPFC) [59] may inflate the reported metabolite levels in participants with SAD. Thus, when reporting metabolites this way, it is essential to determine the levels of tCr in the ROIs being studied.

Several demographic variables also may have potentially contributed to the data heterogeneity. As previously discussed, two studies did not match their SAD and HC participants according to sex and age; thus, study results may have been affected by these covariates. Although evidence indicates that female subjects may be more susceptible to SAD [1], only two studies addressed this issue [54,58]. Consequently, the moderating effect of sex on metabolites needs to be more thoroughly investigated. Furthermore, three studies exclusively recruited those with SAD [55,59,60], whereas the remaining enrolled participants with other comorbidities [53,54,56,57,58]. The effects of other psychiatric disorders on brain metabolite levels may have further confounded the study findings. For instance, ^1^H MRS studies of MDD indicate lower mI in the dmPFC and ACC [95,96]. Henceforth, including SAD with comorbid MDD may have confounded the mI results in the SAD group in some studies [56,58].

The most apparent limitation is that a small number of participants were enrolled in the reviewed studies, affecting adequate statistical comparisons. Generally, a small sample size increases Type II error while decreasing the statistical power. Inadequate sample sizes were particularly problematic for smaller subcortical regions (i.e., caudate, putamen, and thalamus) [55,59,60], for which sample sizes of ≥50 participants per group were needed for adequate comparison. However, studying these regions at higher magnetic fields (≥7T) (providing higher SNR) may decrease the necessity of using the larger samples.

Another apparent limitation is that not all metabolites have been adequately studied. Given that Glu and GABA play an essential role in E/I neurocircuitry in SAD, other regions implicated in the ‘fear neurocircuitry,’ such as dlPFC, insula, the ventral striatum (caudate and putamen), and OFC, should be investigated. Furthermore, the biomarkers of oxidative stress, such as glutathione, taurine, ascorbate, and glycolysis byproducts such as lactate, should be examined in the future ^1^H MRS studies.

Lastly, the effects of treatments were only investigated in two studies, making it challenging to make conclusions about the impact of administered treatments on SAD pathogenesis. Consequently, longitudinal, spectroscopic studies with larger sample sizes are needed to understand the effects of these treatments. Nevertheless, overall results from the eight ^1^H MRS studies indicate that specific molecular mechanisms may be implicated in SAD despite the number of technical and demographic limitations.

## 4. Conclusions

The eight ^1^H MRS studies reviewed in this article suggest regional abnormalities in ‘fear neurocircuitry’ in participants with SAD. The implicated regions include the thalamus, dmPFC, insula, ACC, dlPFC, and subcortical regions that include caudate and putamen. Alterations in NAA, tCho, tCr, mI, GABA, Glu, and Gln were noted.

Despite several demographic, technical, and sample size limitations, the evidence from the eight studies points towards the pathophysiological mechanisms involving the injury to neurons and glial cells. Cellular damage may have resulted from impairments in mitochondrial function, ATP production disturbances, macromolecules essential for maintaining cellular membranes, disturbances in G-protein-coupled second messenger systems, and imbalances in Glu–Gln and Glu–GABA cycling.

The evidence also implies that oxidative stress and glycolysis may also play a role in the pathogenesis of SAD, warranting further ^1^H MRS investigations. The imbalances in ‘fear neurocircuitry’ are also implicated based on the studies examining Glu, Gln, and GABA; however, the contributions of other metabolites (such as NAA, Cho, Cr, and mI) to E/I imbalances should be studied in the future.

More cross-sectional studies with adequate sample sizes are needed to verify the results described in this review, including studies looking at the effects of SAD on GABA, Glu/Gln, lactate and GSH, taurine, or ascorbate. Future research can also benefit from longitudinal spectroscopic studies investigating the effects of pharmacological treatments on neurochemical changes and molecular mechanisms of SAD pathogenesis.

## Figures and Tables

**Figure 1 ijms-23-04754-f001:**
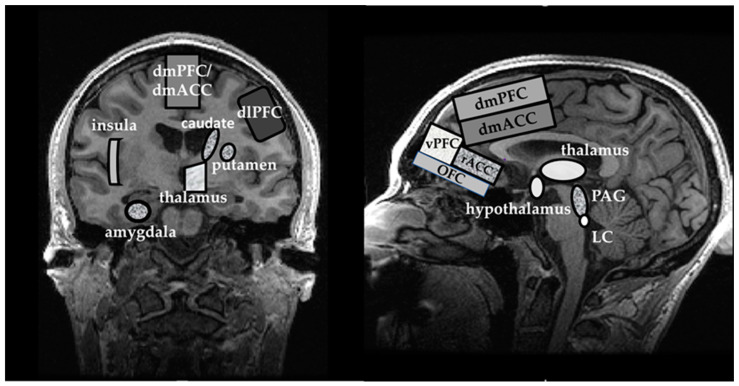
Anatomical structures involved in the ‘fear neurocircuitry’: Coronal view (**left**), Sagittal view (**right**). Abbreviations: dmPFC, dorsomedial Prefrontal Cortex; dmACC, dorsomedial Anterior Cingulate Cortex; dlPFC, dorsolateral Prefrontal Cortex; vPFC, ventral Prefrontal Cortex; rACC, rostral Anterior Cingulate Cortex; OFC, Orbitofrontal Cortex; PAG, Periaqueductal Gray; LC, Locus Coeruleus.

**Figure 2 ijms-23-04754-f002:**
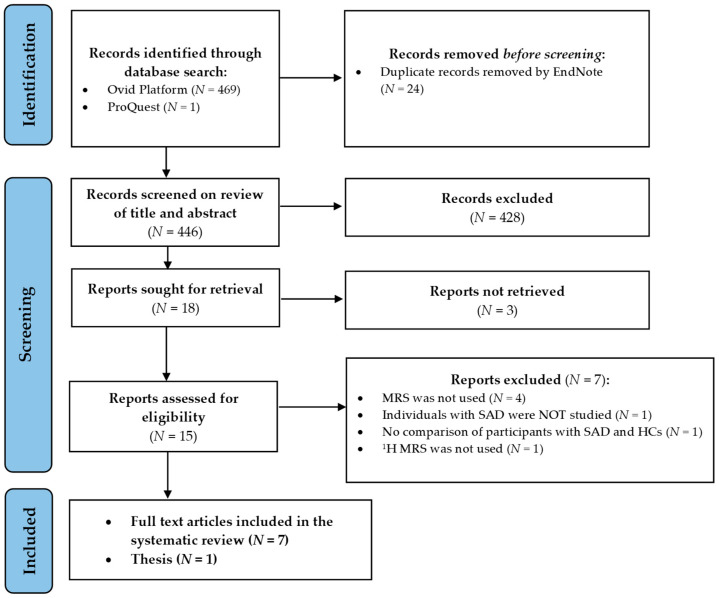
Retrieval flowchart indicating how the data were obtained for systematic review. Abbreviations: *N*, number of publications; MRS, Magnetic Resonance Spectroscopy; SAD, Social Anxiety Disorder; HC, healthy controls; ^1^H MRS, Proton Magnetic Resonance Spectroscopy.

**Figure 3 ijms-23-04754-f003:**
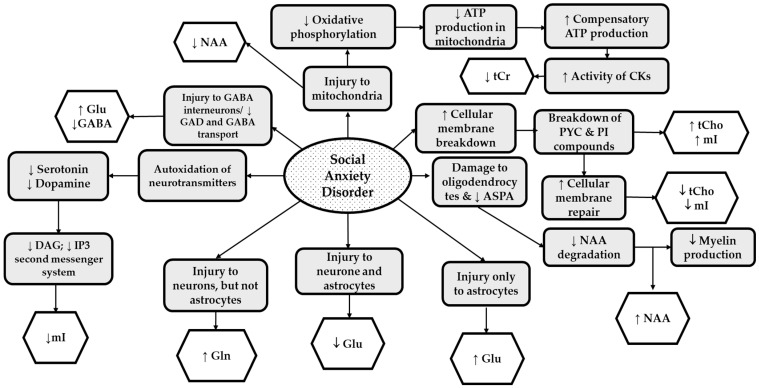
The Summary of Neurochemical Changes Associated with Social Anxiety Disorder: Mechanisms proposed in the ^1^H MRS studies (**light gray boxes**); Reported metabolic changes (**white boxes**). Abbreviations: NAA, N-acetyl aspartate; tCr, total creatine; tCho, total choline; mI, myo-inositol; GABA, gamma-aminobutyric acid; Glu, glutamate; Gln, glutamine; GAD, glutamic acid decarboxylase; ATP, adenosine triphosphate; CK, creatine kinase; PYC, phosphatidylcholine; PI, phosphoinositol; ASPA, aspartoacylase; DAG; diacylglycerol; IP_3_, inositol triphosphate.

**Table 1 ijms-23-04754-t001:** **Study demographic characteristics and ^1^H MRS methods**. Abbreviations: *N*, number of participants; F, females; #, number; SD, Standard Deviation; ^1^H MRS, Proton Magnetic Resonance Spectroscopy; ROI, region of interest; mm^3^, cubic millimeters; T, Tesla; LSAS, Liebowitz Social Anxiety Scale; BSPS, Brief Social Phobia Scale; SAD, Social Anxiety Disorder; HC, Healthy Controls; PTSD, Post-Traumatic Stress Disorder; PD, Panic Disorder; MDD, Major Depressive Disorder; AD, Adjustment Disorder; MDE, Major Depressive Episode; DD, Dysthymic Disorder; SP, Specific Phobia; APD, Avoidant Personality Disorder; NAA metabolite, N-acetyl aspartate metabolite; tCr, total creatine metabolite; tCho, total choline; mI, myo-inositol; GABA, gamma-aminobutyric acid; Glu, glutamate; Gln, glutamine; Glx, glutamix; H_2_O, water; ms, milliseconds; 2D CSI, two-dimensional chemical shift imaging; CGM, cortical gray matter; SCGM, subcortical gray matter; WM, white matter; NCGM, non-cortical gray matter; V, ventricle; ACC, anterior cingulate cortex; dmPFC, dorsomedial prefrontal cortex; dlPFC, dorsolateral prefrontal cortex; OC, occipital cortex; TE, echo time; TR, repetition time; TM, mixed time; STEAM, Stimulated Echo Acquisition Mode; PRESS, Point Resolved Spectroscopy; DQF-S, Double-Quantum and Filter-Selective Editing Technique for Refocusing of the GABA Signal; SNR, signal-to-noise ratios.

Study, Magnetic Field, and Scanner Brand	Participants (*N*; # females; Mean Age ± SD; Age–Sex Matching)	LSAS/BSPS Scores (Mean ± SD)	Psychiatric Comorbidities	Medication Status	^1^H MRS Method; Sequence and Sequence Parameters	Studied, Voxel Size (mm^3^) and Reported Neuro-Metabolites
[55] 1.5 T Philips Medical Systems	SAD: 24; 12F, 28.5 ± 6.63 HC: 24; 12F; 28.38 ± 5.84 Age–sex matched Right-handed only	SAD: 74.04 ± 27.39 HC: Not reported	None reported	Medication-free 6 weeks prior to study enrollment	SVS; STEAM TR = 2500 ms TE = 30 ms TM not reported	Left ACC (2169); Left caudate (1000), Left putamen (1400), Left insula (1920) All voxels: NAA/tCr; tCho/tCr; mI/tCr
[60] 3 T Allergra Siemens	SAD: 18; 11F; 31 ± 9.89 HC: 19; 8F; 29.2 ± 8.15 Age–sex matched	SAD: 88.6 ± 24.82 HC: 21.55 ± 21.19	None reported	Medication-free during the study	SVS for ACC; PRESS TR = 1500 ms TE = 30 ms 2D CSI rest; PRESS TR = 2000 ms TE = 30 ms	ACC (4000) NAA met/Cr; NAA/tCr; Glx/tCr; Glu/tCr, Cho/tCr; Cho/tCr; mI/tCr Bilateral caudate putamen and thalamus (800) NAA met./tCr; Cho/tCr
[59] 3T Philips Achieva	SAD: 9; 4F; 21.6 ± 2.5 HC: 9′ 4F; 21.2 ± 2.0 Age–sex matched	SAD: 57.3 ± 11.5 HC: 26.7 ± 6.0	None reported	Medication-free during the study	SVS; STEAM TR = 2000 ms TE = 20 ms	Left dlPFC (3500), Left ACC (3400), Left putamen (2800), Right putamen (2900), Left thalamus (3200) All regions NAA/H_2_O; Cho/H_2_O; Cr/H_2_O; NAA/tCr; Cho/tCr
[58] 3T Magnex Scientific	Two group comparison All SAD: 36; 19F; 29.86 ± 8.80 HC: 75; 55F; 31.49 ± 9.68 Age–sex not matched Three group comparison SAD: 15; 7F; 28.87 ± 10.13 SAD + MDD: 21; 13F30.57 ± 7.89 HC: 75; 55F; 31.49 ± 9.68 Age–sex not matched	Two-group comparison Median for SAD 75.50 Median for HC 12 Three-group comparison Means ± SD or medians not reported	Two-group comparison None specifically reported Three-group comparison None specifically reported	Two-group comparison Medication-free 4 weeks before enrollment Three-group comparison Medication-free 4 weeks prior to enrollment	All SVS; PRESS For Glu TR = 2400 ms TE = 130 ms For NAA and mI TR = 2400 ms TE = 132 ms PRESS with DQF-S For GABA TR = 2400 ms TE = 130 ms	dmPFC/ACC (2250) GABA/H_2_O Glu/H_2_O NAA/H_2_O mI/H_2_O
[57] 4T Varian, Unity-INOVA	SAD: 10; 2F; 37.2 ± 11.8 HC: 9; 2F; 33.2 ± 11.6 Age–sex matched	SAD: 81.4 ± 19.4 HC: Not reported	GAD: 3 Past MDD: 5 Past alcohol abuse: 5	Medication-free at least 2 weeks prior to enrollment	2D-CSI; MEGA-PRESS with J-editing TR = 1400 ms TE = 30–490 ms	Thalamus (1600) Whole brain All regions GABA/tCr; Glu/tCr; Gln/tCr
[56] 4T Siemens Medical Systems	SAD: 10; 5F; 26.7 ± 6.8 HC: 10; 5F; 26.6 ± 6.8 Age–sex matched Right-handed only	SAD: 72.1 ± 20.6 HC: 9.8 ± 8.9	AD with depressed mood: 1 Past MDE: 1	Medication-free and naive	SVS; STEAM TR = 2000 ms TE = 10 ms TM = 10 ms	ACC (800); OC (800) All voxels tCr; Glu/tCr; NAA/tCr; Cho/tCr; mI/tCr
[54] 1.5T General Electric	SAD: 19; 14F; 42.0 ± 11.6 HC: 10; 4F; 37.8 ± 10.5 Age-sex not matched	LSAS scores not reported	DD: 4 SP: 4 DD and SP: 1	Medication-free at least 2 weeks prior enrollment	2D-CSI; STEAM TR = 1500 ms TE = 20 ms TM = 26 ms	CGM; SCGM; WM All voxels Cho/H_2_O; NAA/H_2_O; Cho/tCr; NAA/tCr; mI/tCr; NAA/tCho; mI/Cho; mI/NAA
[53] 1.5T General Electric	SAD: 20; 9F; 35.7 ± 6.7 HC: 20; 10F; 34.6 ± 9.1 Age–sex matched	LSAS not administered Means ± SDs not reported for BSPS	APD: 8 SP: 4 DD: 4 MDD: 1	Medication-free at least 2 weeks prior to the study enrollment	2D-CSI; STEAM TR = 2000 ms TE = 270 ms TM = 10.6 ms	Voxel 63 (3000): 75% WM, 10% NCGM; 15% V (mostly WM) Voxel 64 (3000): 70% NCGM, 30% WM (WM + NCGM; mostly thalamus) Voxel 65 (3000): 60% NCGM, 30% WM, 10% CGM (mostly GM, including caudate) All regions: tCr SNR; Cho SNR, NAA SNR; NAA/tCr, NAA/Cho

**Table 2 ijms-23-04754-t002:** **Proton Magnetic Resonance Spectroscopy findings from cross-sectional studies**. Abbreviations: ROI, region of interest; *N*, number of participants; SD, Standard Deviation; SAD, Social Anxiety Disorder; HC, Healthy Controls; M, male; F, female; Stat., statistics used; n/a, data not available; NAA, N-acetyl aspartate; tCr, total creatine metabolite; tCho, total choline; mI, myo-inositol; GABA, gamma-aminobutyric acid; Glu, glutamate; Gln, glutamine; Glx, glutamix; H_2_O, water; WM, white matter; GM, gray matter; NCGM, non-cortical gray matter; ACC, anterior cingulate cortex; dmPFC, dorsomedial pre-frontal cortex; dlPFC, dorsolateral prefrontal cortex; OC, occipital cortex; SNR, signal-to-noise ratio; ↑, higher; ↓, downregulated; ↔, no differences between groups; asterisk indicates two-group comparison. NAA metabolite included NAA + NAAG (N-acetylaspartylglutamate) + NAA-N-acetyl-aspartate; Choline metabolites included Cho = glycerophosphacholine (GPC) + phosphocholine; *, asterisk indicates *p*-values less than or equal to 0.05.

ROI	Study	Metabolite	SAD Group			HC Group			Stat.	*p*-Value	Results
			Mean	SD	*N*	Mean	SD	*N*			
NAA											
ACC	[55]	NAA/tCr	1.86	0.05	24	1.80	0.04	24	t = 4.48	<0.001 *	↑ in SAD group
	[60]	NAA met/tCr	1.20	0.07	18	1.23	0.10	18	n/a	n/a	↔
		NAA/tCr	1.09	0.07	18	1.13	0.09	18	n/a	n/a	↔
	[59]	NAA/tCr	1.034	0.184	9	1.019	0.159	9	t = 0.176	0.863	↔
		NAA/H_2_O	14.84	2.081	9	15.173	3.564	9	t = −0.23	0.817	↔
	[56]	NAA/tCr	1.62	0.22	10	1.44	0.15	10	t = 2.19	0.04 *	↑ in SAD group
dmPFC/ACC	[58]	NAA/H_2_O M	9.08	1.11	17	8.5	0.89	20	F = 1.15	0.3	↔
		NAA/H_2_O F	8.77	1.38	19	9.97	1.4	55	F = 4.81	0.03 *	↓ in SAD group
OC	[56]	NAA/tCr	1.06	0.09	10	1.15	0.14	10	t = −2.17	0.04 *	↓ in SAD group
Left insula	[55]	NAA/tCr	1.84	0.05	24	1.8	0.03	24	t = 3.07	0.004 *	↑ in SAD group
dlPFC	[59]	NAA/tCr	1.398	0.16	9	1.23	0.167	9	t = 2.186	0.044 *	↑ in SAD group
		NAA/H_2_O	15.573	1.571	9	16.196	1.56	9	t = −0.842	0.412	↔.
Cortical GM	[54]	NAA/tCr	1.331	0.081	19	1.373	0.097	10	n/a	n/a	↔
Left caudate	[55]	NAA/tCr	1.86	0.03	24	1.85	0.04	24	t = 1.34	0.19	↔
	[59]	NAA/tCr	0.97	0.33	14	1.01	0.2	16	n/a	n/a	↔
Right caudate	[60]	NAA/tCr	1.1	0.52	14	0.98	0.17	16	n/a	n/a	↔
NCGM + WM	[53]	NAA SNR	9.69	6.19	12	23.29	7.2	13	z = 3.67	0.0002 *	↓ in SAD group
		NAA/tCr	1.78	0.45	12	2.12	0.49	13	t = 1.95	0.06	↔
Subcortical GM	[54]	NAA/tCr	1.228	0.094	19	1.276	0.082	10	n/a	n/a	↔.
Left putamen	[55]	NAA/tCr	1.86	0.03	24	1.85	0.04	24	t = 1.34	0.19	↔
	[60]	NAA/tCr	0.97	0.33	14	1.01	0.2	16	n/a	n/a	↔
	[59]	NAA/tCr	0.987	0.158	9	0.95	0.207	9	t = 0.438	0.667	↔
		NAA/H_2_O	13.398	1.552	9	12.871	1.612	9	t = 0.719	0.482	↔
Right putamen	[60]	NAA/tCr	1.02	0.21	17	1.06	0.16	18	n/a	n/a	↔
	[59]	NAA/tCr	1.02	0.21	17	1.06	0.16	18	t = 0.191	0.851	↔
		NAA/H_2_O	12.31	1.786	9	12.809	1.692	9	t = −0.608	0.551	↔
Left thalamus	[60]	NAA/tCr	1.39	0.39	17	1.17	0.22	17	z = 1.92	0.054	↔
	[59]	NAA/tCr	1.475	0.296	9	1.362	0.399	9	t = 0.715	0.484	↔
		NAA/H_2_O	14.96	2.404	9	14.209	4.071	9	t = 0.506	0.62	↔
Right thalamus	[60]	NAA/tCr	1.49	0.39	17	1.21	0.25	17	z = 2.14	0.031 *	↑ in SAD group
Mostly NCGM	[53]	NAA SNR	15.68	7.04	20	26.39	10.78	19	z = 3.16	0.001 *	↓ in SAD group
Mostly WM	[53]	NAA SNR	11.14	4.83	20	15.9	5.4	17	z = 2.66	0.0007 *	↓ in SAD group
		NAA/tCr	1.78	0.41	20	1.99	0.32	17	t = 1.72	0.09	↔
		NAA/Cho	1.93	0.53	20	2.26	0.40	17	t = 2.14	0.03 *	↓ in SAD group
WM	[54]	NAA/tCr	1.312	0.119	19	1.368	0.125	10	n/a	n/a	↔
tCho											
ACC	[55]	tCho/tCr	0.83	0.05	24	0.84	0.05	24	t = −0.2	0.84	↔
	[60]	tCho met/tCr	0.28	0.03	18	0.29	0.03	18	n/a	n/a	↔
	[59]	tCho/tCr	0.292	0.08	9	0.253	0.063	9	t = 1.06	0.307	↔
		tCho/H_2_O	4.186	1.057	9	3.71	0.929	9	t = 0.942	0.362	↔
	[56]	tCho/tCr	0.49	0.07	10	0.57	0.09	10	t = −2.19	0.04 *	↓ in SAD group
Left insula	[55]	tCho/tCr	0.77	0.03	24	0.77	0.19	24	t = 0.02	0.99	↔
dlPFC	[59]	tCho/tCr	0.209	0.034	9	0.207	0.034	9	t = 0.149	0.883	↔
		tCho/H_2_O	2.373	0.597	9	2.757	0.657	9	t = −1.296	0.213	↔
Cortical GM	[54]	tCho/tCr	0.876	0.071	19	0.806	0.046	10	n/a	<0.01 *	↑ in SAD group
Left caudate	[55]	tCho/tCr	0.78	0.05	24	0.77	0.04	24	t = 0.5	0.62	↔
	[60]	tCho met/tCr	0.23	0.06	14	0.24	0.04	16	n/a	n/a	↔
Right caudate	[60]	tCho met/tCr	0.22	0.04	14	0.25	0.07	16	n/a	n/a	↔
NCGM + WM	[53]	tCho SNR	6.24	4.48	12	12.5	3.64	13	z = 3.18	0.001 *	↓ in SAD group
Subcortical GM	[54]	tCho/tCr	0.882	0.055	19	0.845	0.058	10	n/a	<0.1	↔
Left putamen	[55]	tCho/tCr	0.87	0.03	24	0.86	0.06	24	t = 0.44	0.66	↔
	[60]	tCho met/tCr	0.25	0.06	17	0.26	0.03	18	n/a	n/a	↔
	[59]	tCho/tCr	0.191	0.038	9	0.216	0.027	9	t = −1.595	0.129	↔
		tCho/H_2_O	2.596	0.483	9	3.006	0.664	9	t = −1.565	0.136	↔
Right putamen	[60]	tCho met/tCr	0.23	0.06	17	0.28	0.03	18	t = −2.86	0.0042 *	↓ in SAD group
	[59]	tCho/tCr	0.154	0.078	9	0.184	0.047	9	t = −0.977	0.343	↔
		tCho/H_2_O	2.164	1.32	9	2.557	0.735	9	t = −0.779	0.447	↔
Left thalamus	[60]	tCho met/tCr	0.29	0.04	17	0.29	0.04	17	n/a	n/a	↔
	[59]	tCho/tCr	0.269	0.08	9	0.215	0.073	9	t = 1.489	0.155	↔
		tCho/H_2_O	2.761	0.84	9	2.293	0.792	9	t = 1.228	0.236	↔
Right thalamus	[60]	tCho met/tCr	0.3	0.06	17	0.28	0.05	17	n/a	n/a	↔
Mostly NCGM	[53]	tCho SNR	8.53	3.18	20	13.29	5.15	19	z = 3.02	0.002 *	↓ in SAD group
WM	[54]	tCho/tCr	0.928	0.065	19	0.924	0.089	10	n/a	n/a	↔
Mostly WM	[53]	tCho SNR	5.79	2.51	20	6.79	2.94	17	n/a	n/a	↔
mI											
ACC	[55]	mI/tCr	0.31	0.04	24	0.33	0.05	24	t = −1.31	0.2	↔
	[60]	mI/tCr	0.98	0.1	18	0.98	0.09	18	n/a	n/a	↔
dmPFC/ACC	[58]	mI/H_2_O	4.54	0.62	36	5.25	0.97	75	t = 3.64	0.001 *	↓ in SAD group
Left insula	[55]	mI/tC	0.26	0.03	24	0.28	0.04	24	t = −1.65	0.11	↔
Cortical GM	[54]	mI/tC	0.994	0.089	19	0.887	0.093	10	n/a	<0.01 *	↑ in SAD group
Left caudate	[55]	mI/tC	0.35	0.03	24	0.36	0.05	24	t = −1.14	0.26	↔
Left putamen	[55]	mI/tC	0.35	0.03	24	0.33	0.04	24	t = 2.51	0.16	↔
Subcortical GM	[54]	mI/tC	0.982	0.113	19	0.89	0.101	10	n/a	<0.05 *	↑ in SAD group
WM	[54]	mI/tC	1.031	0.097	19	0.989	0.073	10	n/a	n/a	↔
tCr											
ACC	[59]	tCr	14.737	3.223	9	14.836	2.276	9	t = −0.069	0.946	↔
	[56]	tCr	4.18	0.5	10	4.38	0.61	10	t = −0.8	0.43	↔
dlPFC	[59]	tCr	11.217	1.297	9	13.392	2.22	9	t = −2.539	0.022 *	↓ in SAD group
OC	[56]	tCr	4.39	0.21	10	4.43	0.42	10	t = −0.31	0.76	↔
Left putamen	[59]	tCr	13.73	1.63	9	13.834	1.769	9	t = −0.132	0.896	↔
Right putamen	[59]	tCr	13.303	2.234	9	13.913	2.063	9	t = −0.602	0.556	↔
NCGM + WM	[53]	tCr SNR	5.59	3.97	12	11.23	4.08	13	z = 3.07	0.001 *	↓ in SAD group
Left thalamus	[59]	tCr	10.402	2.136	9	11.053	3.687	9	t = −0.487	0.633	↔
NCGM + WM	[53]	tCr SNR	7.64	2.98	20	12.78	3.94	19	z = 3.75	0.0002 *	↓ in SAD group
Mostly WM	[53]	tCr SNR	6.37	2.8	20	7.93	3.4	17	n/a	n/a	↔
Glu											
ACC	[60]	Glu/tCr	1.62	0.193	18	1.76	0.18	18	F = 5.07	0.031 *	↓ in SAD group
	[56]	Glu/tCr	1.37	0.18	10	1.21	0.11	10	t = 2.39	0.03 *	↑ in SAD group
dmPFC/ACC	[58] M	Glu/H_2_O	11.98	1.42	17	10.48	1.3	20	t = −2.61	0.02 *	↑ in SAD group
	[58] F	Glu/H_2_O	11.17	1.03	19	11.33	1.4	55	t = 0.44	0.66	↔
OC	[56]	Glu/tCr	1.07	0.09	10	1.16	0.14	10	t = −1.69	0.11	↔
Thalamus	[57]	Glu/tCr	1.07	0.22	10	0.91	0.22	10	t = −1.41	0.18	↔
The Whole Brain	[57]	Glu/tCr	1.37	0.43	10	0.99	0.21	10	t = −2.22	0.04 *	↑ in SAD group
tGln											
Thalamus	[57]	Gln/tCr	0.43	0.18	10	0.19	0.06	10	t = −3.24	0.008 *	↑ in SAD group
The Whole Brain	[57]	Gln/tCr	0.57	0.33	10	0.23	0.06	10	t = −2.88	0.01 *	↑ in SAD group
tGlx											
ACC	[60]	Glx/tCr	2.01	0.27	18	2.17	0.33	18	n/a	n/a	↔
GABA											
dmPFC/ACC	[58]	GABA/H_2_O	0.97	0.26	36	1.1	0.24	75	t = 2.58	0.01 *	↓ in SAD group
Thalamus	[57]	GABA/tCr	0.05	0.02	10	0.12	0.07	10	t = 2.17	0.05 *	↓ in SAD group
The Whole Brain	[57]	GABA/tCr	0.09	0.09	10	0.11	0.07	10	t = 0.42	0.69	↔

## Data Availability

Not applicable.

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
