# Peer review of "Neurochemical Alterations in Social Anxiety Disorder (SAD): A Systematic Review of Proton Magnetic Resonance Spectroscopic Studies"

_ijms, 2022, doi:10.3390/ijms23094754_

Round 1

Reviewer 1 Report

The first sentence listed SAD as the 4th most prevalence psychiatric disorder, but the other 3 that come before it should be stated to complete the comparative context provide.

Figure 1. should show in color green for excitatory and red for inhibitory pathways with lines/arrows suggesting how these circuits work and their directionality. This would make the point much clearer.

Sections 1.1, 1.3., 1.5., and 1.6 should be all one paragraph and not spit into two mini-paragraphs.

In Figure 2. all the "n" should be italicized throughout. And technically, the indentification of sample papers should be "N" and not "n" as the represent that full sample from which, then smaller "n" samples were taken.

For Table 2. the statistical p-value column should have the asterisk following the p-values to visually allow the reader to discern relatively quickly which values were significant or not.

Same issue. In all the results section, make the sub-heading sections all one paragraph and not split mini-paragraphs.

Figure 3 is excellent!

Otherwise, this was a very enjoyable and interesting read. The article was well-written and clear.

Reviewer 2 Report

The manuscript from Elsaid and colleagues is very interesting and brings novelty through a systematic review of H MRS studies in SAD individuals. I have a few suggestions and I greatly suggest to the authors explore the results through a metanalysis test.

Introduction:

Page 53: Please, identify "glu" in the first time no text, not in line 90.

Page 117: I am not sure metastasis is an example of mechanism of cellular membrane repair.

Page 126: I suggest presenting the concentration as mM as in the other sections.

Methods

Which Boolean operators did the authors used?

page 217: " During the screening process, addictional 428 articles were excluded
subsequently excluded." Please, correct this sentence.

Results

Figure 2: "1H MRS, proton magnetic resonance spectroscop"... the letter "y" is missing.

Is it possible to realize a meta-analysis test? I think your data met the criteria for this test.

Page 326: I think the second "Davidson" in the sentence is a typographical mistake.
